# Marital Stability and Quality of Couple Relationships after Acquired Brain Injury: A Two-Year Follow-Up Clinical Study

**DOI:** 10.3390/healthcare9030283

**Published:** 2021-03-04

**Authors:** Stefania Laratta, Lucia Giannotti, Paolo Tonin, Rocco Salvatore Calabrò, Antonio Cerasa

**Affiliations:** 1S. Anna Institute, 88900 Crotone, Italy; s.laratta@isakr.it (S.L.); lucia.giannotti@libero.it (L.G.); p.tonin@isakr.it (P.T.); 2IRCCS Centro Neurolesi Bonino-Pulejo, 98124 Messina, Italy; roccos.calabro@irccsme.it; 3Institute for Biomedical Research and Innovation, National Research Council (IRIB-CNR), 87050 Mangone, Italy

**Keywords:** marital stability, acquired brain injury, spouses, religiosity, educational level

## Abstract

Couple relationships after acquired brain injury (ABI) could be vulnerable to emotional distress. Previous evidence has demonstrated significant marital dissatisfaction in the first period after a traumatic event, while long-term evaluations are lacking. In this study, we evaluated the impact of a series of demographic and clinical factors on marital stability after two years from the injury. Thirty-five patients (29% female) with mild/moderate ABI (57% vascular, 43% traumatic) and their partners were enrolled. The couples completed a series of psychological questionnaires assessing marital adjustment (Dyadic Adjustment Scale, DAS) and family functioning (Family Relationship Index, FRI) at discharge from the intensive rehabilitation unit and after 2 years. Demographics (i.e., educational level, job employment and religion commitment) and clinical variables (i.e., the Barthel index, aetiology and brain lesion localization) were considered as predictive factors. Regression analyses revealed that the DAS and FRI values are differently influenced by demographic and clinical factors in patients and caregivers. Indeed, the highest educational level corresponds to better DAS and FRI values for patients. In the spouses, the variability of the DAS values was explained by aetiology (the spouses of traumatic ABI patients had worse DAS values), whereas the variability in the FRI values was explained by religious commitment (spending much time on religious activities was associated with better FRI values). Our data suggest that some clinical and demographic variables might be important for protecting against marital dissatisfaction after an ABI.

## 1. Introduction

Acquired brain injury (ABI) determines cognitive, emotional, and behavioural changes negatively affecting the patient’s feelings and family relationships. These could be an important cause of stress and marital dissatisfaction. Studies on couple satisfaction after a traumatic brain injury (TBI) or stroke have, in fact, documented a decline in marital stability and psychological health [1,2,3,4]. Assisting a person with cognitive disabilities as a result of brain damage could transform the relationship from reciprocal to unilateral, causing a stressful change in couples that makes it easier for depressive symptoms to develop in the spouses [5,6] or leads to a high incidence of divorce [7]. 

It has been demonstrated that the spouses’ quality of life is negatively influenced by brain pathological events, given that the rate of post-event separation and divorce may increase by up to 78% [2,8]. The spouse often starts to neglect the couple relationship in favour of a care relationship in which he/she takes care only of the daily life of the partner (washing, dressing or bathing) and the management of his/her health status (physiotherapy, medication and medical examinations) [9]. Moreover, the caregiver is also engaged in taking full charge of children, house and work, whereas love comes to be replaced by other feelings mostly associated with the role of protection and care [10]. Another central construct of dyadic cohesion, sexual intimacy, may not be the same, likely due to sexual dysfunctions or other physical/behavioural problems that interfere with an intimate relationship [11].

However, chronic disability is not incompatible with a good quality of life for couples, which has been considered a protective factor for a better prognosis in chronic diseases [12], such as Parkinson’s disease [13], heart disease [14] and cancer [15]. Couple stability is highly dependent on the stress level as perceived by the caregiver [16]. Quinn et al. [17] perfectly described the real difficulty in post-stroke couples, where caregivers feel as if they are living with a stranger, having lost the dialogue and sharing of problems. Generally, the spouses suffer from no longer being supported by their partners and feel, however, a relational obligation to take care of them [17]. Similarly, in TBI couples, caregivers describe irritability and aggression, anxiety and social isolation as common problems of their lives [18]. 

Unlike most of the studies in the literature, focused mainly on the psychological aspects of spouses, in our study, we were also interested in the concept of couple stability, which expresses the survival of the couple relationship in relation to the degree of conjugal satisfaction [17]. Marital stability has been investigated and classified, labelling couples as married, separated or divorced [19]. Researchers have suggested that postinjury marital relationships are prone to instability and divorce in comparison to the general population. Nonetheless, some authors indicate relatively high levels of marital stability despite high levels of marital distress [8]. Moreover, it has been shown that among those who are single/divorced/separated after ABI, 87% remained so at two-year follow-up [2], whereas 13% underwent positive change [4]. 

However, the relationship between a patient affected by ABI and his/her spouse is more complex and depends on many variables, including age, schooling, social background and premorbid conditions. Notably, injury during deployment is believed to significantly predict positive relationship change [4]. Moreover, changes in spousal perceptions, interactions, responsibilities and reactions to brain injury may affect marital stability and satisfaction [3].

The rationale of this paper stems from the need to better clarify the variables subtending marital stability after ABI. To this end, we sought to analyse how the perception of a marital relationship is modulated by a series of clinical (i.e., aetiology and clinical status) and demographic variables (the duration of the marriage, presence or absence of children, life cycle and influence of religiosity), taking into account the points of view of both spouses and patients. 

## 2. Materials and Methods

### 2.1. Participants

The study involved couples in which one person was affected by a traumatic or vascular ABI. Patients consecutively admitted to the Intensive Rehabilitation Unit (IRU) of the Institute S. Anna (Crotone, Italy) between January 2013 and December 2019 were screened for possible inclusion. We only considered patients who were married at the time of enrolment and for the entire follow-up period. The inclusion criteria were (i) an age range of 18 to 70 years, and (ii) patients who had completed the rehabilitation process without severe cognitive deficits (i.e., with a Montreal Cognitive Assessment score >25) at the discharge. The exclusion criteria included (a) the presence of a premorbid history of psychiatric diseases, and (b) being divorced or separated. 

After the initial screening, we selected 101 ABI patients. Twenty-eight refused to participate, 17 did not return the questionnaires, and 17 died before follow-up evaluation. The remaining 39 former pairs of patients and spouses were contacted at 2-year follow-ups (see Figure 1). Thirty-five couples (29% female) concluded the study, because, during the follow-up period, four couples divorced (10%).

All the patients and their spouses gave written informed consent, and the study was approved by the Ethical Committee of the Central Area Regione Calabria of Catanzaro, according to the Helsinki Declaration.

### 2.2. Design and Procedure 

All of the included couples were evaluated at discharge from the IRU and after two years from the injuries. The follow-up evaluation was conducted by telephone, and the patients were asked not to be helped by their partners (or others) to complete the questionnaires. All the participants completed a demographic data collection form, including information on age, educational level, job employment before the traumatic event, the duration of the relationship (years), the number of children, the family life cycle (initial engagement/spouse without child/spouse with young child/spouse with adult child), spirituality and the frequency of participation in religious activities. For the clinical variables, we collected data regarding the Barthel Index and ABI aetiology (vascular and traumatic).

### 2.3. Outcome Measures

The quality of the couple relationships was evaluated by means of the Dyadic Adjustment Scale (DAS [20]) and the Family Relationship Index (FRI [21]).

The DAS is one of the most widely used tools in the clinic and research to evaluate the functioning of a couple [22]. It is a simple, self-administrable and fast compilation scale composed of 32 items rated on a 5-point Likert scale and divided into four subscales: couple satisfaction, dyadic consent, couple cohesion and affective expression. The couple satisfaction subscale consists of 15 items and assesses the happiness or unhappiness that couples perceive with respect to their relationship; the dyadic consent scale, composed of seven items, observes the frequency of disputes, and the pleasure or otherwise of being together, taking into account separation or divorce. The dyadic cohesion scale is composed of five items and assesses the amount of time in which partners share pleasant activities such as social interests, dialogue or having common goals. Finally, the emotional expression scale consists of four items and assesses how the couple express their feelings, love and sexuality. The sum of the four scales provides a total score that expresses the couple’s overall degree of agreement. Total Couple Adaptation has a theoretical range from 0 to 151, corresponding to the minimum and maximum sums of all the items. The theoretical construct is based on the fact that the coupling adjustment, or, generically, relationship quality, can be understood both as an individual property (the perception of individual feeling) and as a dyadic index (the perceptions of the feelings of a couple). 

The FRI scale assesses the quality of a family in a marital relationship. It provides a global index and three partial values relating to three subscales: family cohesion (family commitment, help and support are provided to each other), communication (expressing feelings directly) and conflict (the management of expressed anger). The FRI consists of 12 items (4 per subset) asking questions (yes/no) about family relationships. A total score of 9 or less (the number of points from “Yes” answers) is indicative of problematic family relationships.

### 2.4. Statistical Analysis

Statistical analysis was performed using the Statistical Package for Social Science software (SPSS, v20.0, Chicago, IL, USA) for Macintosh. Assumptions of normality were tested for all the continuous variables. Normality was tested using the Kolmogorov–Smirnov test. A paired t-test was used to evaluate the differences in demographic factors within the couples, and to evaluate significant changes in the relationship before and after the IRU period. Stepwise multiple regression analysis was performed to evaluate the impact of clinical variables (i.e., the Barthel Index, aetiology, brain lesion localization and side of the lesion) and demographic variables (age, educational level, job employment, the duration of the relationship, the life cycle, the number of children and religious orientation/commitment) on the quality of the relationship and family (DAS and FRI). For all the tests, a *p*-value < 0.05 was considered to be statistically significant. 

## 3. Results

### 3.1. Clinical Data

The clinical characteristics of the patients are reported in Table 1. At discharge, they were of good clinical status (mean Barthel Index = 61 ± 23.7) with no evidence of unbalanced clinical characteristics (aetiology or hemispheric lesions). 

The demographic characteristics of the patients and their respective spouses are reported in Table 2. An unpaired t-test revealed no significant differences in age (*p* = 0.44) or educational level (*p* = 0.99). The average duration of a relationship was 31.7 ± 13.1 years, with the majority of the participants having children (median value, 2 (0–4)) and similar Christian spiritual orientations and commitments (all *p*-values > 0.68). Considering the life cycles of the couples, 9% were in initial engagements, 17% were spouses without a child, 34% were spouses with a young child, and 40% were spouses with an adult child. 

In evaluating the psychological data at discharge and after 2 years, no significant differences in the patients and spouses as concerned the quality of relationships (DAS; *p*-value = 0.93; *p*-value = 0.64, respectively) and family (FRI; *p*-value = 0.75; *p*-value = 0.74, respectively) were detected.

### 3.2. Regression Analysis

We evaluated if demographic or clinical variables could affect the quality of the couple and family relationships for the enrolled spouses. Considering the DAS as dependent variables, the only variable surviving analysis for the patient group was the educational level. Indeed, the highest educational level (high school/university) corresponded to the highest DAS values (R = 0.49; R^2^ = 0.24; beta = 0.49; *p*-value = 0.007). In the spouses, the variability of the DAS values was, instead, explained by the aetiology. In other words, the spouses of traumatic ABI patients had lower DAS values (R = 0.38; R^2^ = 0.15; beta = −0.38; *p*-value = 0.03). Considering the FRI as dependent variables, similarly to previous findings, the highest educational level (high school/university) corresponded to the highest FRI values only for the patients’ group (R = 0.51; R^2^ = 0.26; beta = 0.51; *p*-value = 0.004). In the spouses, the variability of the FRI values was, instead, explained by religion commitment. The spouses who spent much time on religious activities had higher FRI values (R = 0.38; R^2^ = 0.15; beta = −0.38; *p*-value = 0.03).

## 4. Discussion

We have demonstrated that there are some demographic and clinical factors that could reduce marital instability during the chronic phase in couples with ABI spouses. Indeed, educational level and religious commitment, as well as aetiology, were differently associated with the perceived couple and family satisfaction. The data collected on the perception of the quality of family and couple relationships suggest that both the patients and the spouses do not perceive their relationships to be significantly changed after a destabilizing event such as brain injury. This finding may be explained in several ways. On one hand, the spouse of a patient who has experienced the threat of losing his/her partner generally underestimates the negative aspects of his/her relationship [23]. Again, the patients live in a condition of dependence and gratitude towards their partners, who are appreciated for the time invested in caring [24]. Another factor influencing the detected couple stability could be related to the time interval between the neurological event and our psychological assessment [25]. The follow-up period (two years) after brain injury was chosen according to previous studies [1,25], which defined the first few months of caregiving as taking up the role, a time when caregivers try to gain control of the situation and attempt to understand their new role. It has been proposed that the first 6 months could be considered as an acute phase, and only after this period may the marriage become a new relationship where the caregiver recognizes the patient’s needs and learns how to face them [25]. Indeed, it has been demonstrated 1 year after injury that the average level of the burden decreases over this period [26].

However, there are other factors that may preserve marital stability in couples with ABI spouses.

Our results show that the higher the educational level, the better the DAS and FRI values for ABI patients. Generally, the level of education is considered one of the clearest indicators of life outcomes such as employment, income and social status [27]. Our findings could be explained by the cognitive reserve hypothesis, which posits that intellectual enrichment is associated with cerebral and cognitive efficiency [28]. Patients with a greater cognitive reserve could have a better cognitive recovery and, therefore, experience less impact on their lifestyle and couple relationship. As previously demonstrated by Schneider et al. [29], the preinjury educational attainment is a strong independent predictor of long-term functional outcomes in ABI patients, suggesting that a brain’s “cognitive reserve” may play a role in helping people get back to their previous lives. Alternatively, the detected impact of educational level on the couple stability might be explained by the relationship with the social status. Indeed, we might hypothesize that patients with the highest educational level are characterized by a better economic status. Consequently, a postinjury life in a comfortable social status may positively affect relationships, providing additional tools for addressing issues of daily life associated with ABI [30]. 

Another interesting observation emerging from our study is that the perception of the quality of the relationship by the spouses of patients with traumatic ABI was worse than for patients with vascular aetiology. A previous study suggests that a burden is prevalent among family caregivers of individuals with TBI, significantly impacting wellbeing and quality of life [31]. However, this effect could be related to the age at the disease onset. Indeed, large epidemiological studies have indicated that the vast majority of TBI victims are in adulthood, whereas stroke patients are characterized by older age [32]. This could, in part, explain the lower impact of stroke outcomes on the quality of the couple relationship. After ABI, older people often stop working and probably reduce their social lives, and as a consequence, the couples are consolidated with lower expectations about the future. However, a younger patient with ABI outcomes and his/her spouse may look at life events in a more dramatic way, related to economic difficulties or the loss of a job and hope for the future. Future studies on larger samples are needed to disentangle the different impacts of age at onset and aetiology on the marital stability for ABI patients. 

It is well known that physical and mental health are strongly affected by spirituality and religiosity [33]. Our results highlight the positive influence of religiosity on the perception of family relationships. This finding is in agreement with previous literature [28], suggesting that religious coping is a predictor of positive outcomes regarding stress levels during a tragic event. Corallo et al. [34] demonstrated that caregivers with religious beliefs used avoidance strategies more frequently than a nonbeliever group, probably for a lack of awareness about the patient’s disease and their disability degree. Religiosity can provide hope, optimism, energy, security and inner strength for solving problems [35]. It provides a sense of confidence about the future and appears to be a significant coping resource post-ABI, alleviating the burden of a sudden disability. In addition to the positive effect on patients, the importance of religiosity for the coping of caregivers has also been studied, especially in family members of patients with dementia [36]. An Iranian study indicated that religious and spiritual beliefs have a role in caregiver adaptations to the situation [37]. They found a significant correlation between positive religious coping and caregivers’ psychological wellbeing. The positive effect of religiosity has been described both in the caregivers of stroke survivors [38] and in victims of TBI [39]. Therefore, believing in a superior entity as well as a high frequency of religious commitment may be considered an important factor protecting against marital instability in couples with ABI spouses.

Some aspects might have affected our analysis. Firstly, apart from the relatively small sample size, a selection bias should be considered. A large number of patients fulfilling the inclusion criteria refused to participate. The recruitment of a substantially larger sample would mitigate concerns about the representativeness of our sample, although this type of issue is typical when performing follow-up studies with a wide time window in this type of neurological patient. Secondly, we could not explore the direct causality between the patients’/caregivers’ demographic characteristics and the quality of the relationships. Next, we only selected patients without major comorbidities and pre-existing physical and cognitive deficits. Consequently, these results cannot be generalized to all ABI populations. Finally, it is worth noting that every consideration about the protective effects of marital instability should include the quality of the relationships characterizing couples in the early years of marriage, before the occurrence of neurological disorders. This information could facilitate the better social profiling of successful marital transitions to the assistance of partners with ABI, thus addressing potential couple therapy interventions aimed at potentiating awareness of the impact.

## 5. Conclusions

The present study shows that couples with ABI spouses are able to find new adjustments to protect their relationships. In particular, we describe some demographic and clinical factors that may have influenced the detected couple stability, which need to be taken into account in order to evaluate interventions designed to promote marital satisfaction during the chronic phase of the disease.

## Figures and Tables

**Figure 1 healthcare-09-00283-f001:**
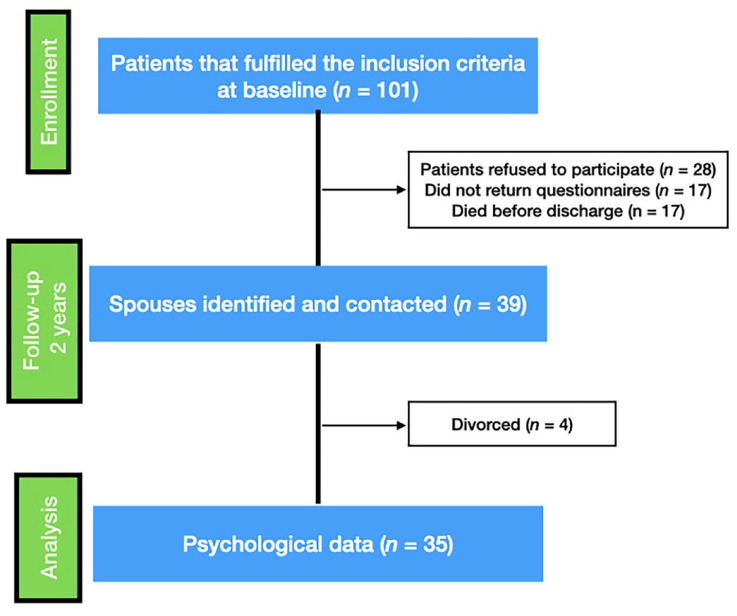
Flow diagram of participant recruitment and participation in the study.

**Table 1 healthcare-09-00283-t001:** Clinical characteristics of acquired brain injury (ABI) patients.

Clinical Variables	Values *(Mean ±SD/Percentage)*
Barthel Index at discharge	61 ± 23.7
ABI phenotype	• 57% Vascular • 43% Traumatic
Brain lesion localization	• 44% Frontal Lobe • 34% Temporal Lobe • 7% Parietal Lobe • 15% Occipital Lobe
Hemispheric lesion (% left)	46% left

**Table 2 healthcare-09-00283-t002:** Demographic and psychological characteristics.

Variables	Patients	Spouses	*p*-Value
*Demographic*			
Age (years)	57.5 ± 1.7	55.7 ± 11.1	0.44 ^§^
Educational level	14% elementary school 32% middle school 37% high school 17% university	11% elementary school 29% middle school 37% high school 23% university	0.22 *
Job employment (Y/N)	71% no	51% no	0.15 *
Spiritual orientation (Y/N)	8% no	5% no	0.56 *
Religious commitment	25% never 28% rarely 34% often 13% very often	15% never 28% rarely 39% often 18% very often	0.22 *
*Psychological*			
FRI at discharge	9.2 ± 2.8	9.4 ± 2.4	0.75 ^§^
FRI after 2 years	9.4 ± 2.7	9.6 ± 2.2	0.74 ^§^
DAS at discharge	109.8 ± 14.1	109.5 ± 16.1	0.93 ^§^
DAS after 2 years	108.6 ± 17.4	106.7 ± 16.9	0.64 ^§^

Data are given as mean values (SDs) or median values (ranges) when appropriate. FRI: Family Relationship Index; DAS: Dyadic Adjustment Scale; * = Chi^2^; ^§^ = Two-sample *t*-test.

## Data Availability

The datasets generated during and/or analysed during the current study are available from the corresponding author on reasonable request.

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
