# Peer review of "Marital Stability and Quality of Couple Relationships after Acquired Brain Injury: A Two-Year Follow-Up Clinical Study"

_healthcare, 2021, doi:10.3390/healthcare9030283_

Round 1

Reviewer 1 Report

Healthcare (MDPI) Review – Laratta et al. 02/15/2021

Comments:

In this work, the authors examine factors that influence marital quality for people who suffered an acquired brain injury (ABI) along with these individuals’ partners. The study examined martial quality in terms of adjustment and family functioning with data collected at two time points, 2 years apart, and controlled for a host of socio-demographic variables.

Overall, I thought the study was presented in a clear way, addresses an important topic relevant to medical and social science disciplines, and used sound methods. The results were thorough with excellent statistical reporting per the standards I use in my field (social sciences). I appreciate the effect sizes being reported with the regression analyses, for example.

The main changes that seem warranted to me are in the limitations. The authors identify two key limitations, which I concur with. However, the study was not entirely able to control for who the participants were and had no data about marital quality prior to the ABI. I understand how studies that have such data would be difficult to conduct due to the ABI events being unpredictable for one. However, this raises the possibility that perhaps the relatively small sample captured data from couples who already had an established strong relationship? This is all the more possible if, through informed consent, couples knew it was a study about marital relationships. In such a case, perhaps the people who did not agree to participate already had a struggling marriage? If so, these data may apply to a unique sample. I do not believe it’s possible to account for this possibility, but to me, it seems worth noting.

Minor comments…

On line 143, “a majority characterized by sons” was not clear to me – is it that the majority of participants with children identified as having sons?

There are some very minor language issues, which is understandable. For example lines 74-75 the exclusion criteria “to be divorced or separated” could be changed to “being divorced or separated” is more grammatically sound. On line 145, “17% spouses without a child” could be better written “17% were a spouse without a child.” These do not detract from the work, but a review for these sorts of minor revisions would enhance the writing.

In sum, I have only minor suggestions for the present manuscript. I appreciate the clear presentation of the background literature, the excellent clarity outlining the analysis plan, and well-presented statistical work. I wish all authors took the time to assure their manuscript submissions had such quality. I think this is a worthwhile manuscript to add to the literature.

Author Response

In this work, the authors examine factors that influence marital quality for people who suffered an acquired brain injury (ABI) along with these individuals’ partners. The study examined martial quality in terms of adjustment and family functioning with data collected at two time points, 2 years apart, and controlled for a host of socio-demographic variables.

Overall, I thought the study was presented in a clear way, addresses an important topic relevant to medical and social science disciplines, and used sound methods. The results were thorough with excellent statistical reporting per the standards I use in my field (social sciences). I appreciate the effect sizes being reported with the regression analyses, for example.

REPLY: We would like to express our appreciation for this reviewer’s revision and comments about our work.

The main changes that seem warranted to me are in the limitations. The authors identify two key limitations, which I concur with. However, the study was not entirely able to control for who the participants were and had no data about marital quality prior to the ABI. I understand how studies that have such data would be difficult to conduct due to the ABI events being unpredictable for one. However, this raises the possibility that perhaps the relatively small sample captured data from couples who already had an established strong relationship? This is all the more possible if, through informed consent, couples knew it was a study about marital relationships. In such a case, perhaps the people who did not agree to participate already had a struggling marriage? If so, these data may apply to a unique sample. I do not believe it’s possible to account for this possibility, but to me, it seems worth noting.

REPLY: Absolutely! We completely agree with this very important suggestion. Obviously, we cannot predict if and how the quality of relationships before the occurrence of ABI may impact marital stability. We now include a new section in the limitations to highlight this point.

Minor comments…

On line 143, “a majority characterized by sons” was not clear to me – is it that the majority of participants with children identified as having sons?

REPLY: Done

There are some very minor language issues, which is understandable. For example lines 74-75 the exclusion criteria “to be divorced or separated” could be changed to “being divorced or separated” is more grammatically sound. On line 145, “17% spouses without a child” could be better written “17% were a spouse without a child.” These do not detract from the work, but a review for these sorts of minor revisions would enhance the writing.

REPLY: Done

In sum, I have only minor suggestions for the present manuscript. I appreciate the clear presentation of the background literature, the excellent clarity outlining the analysis plan, and well-presented statistical work. I wish all authors took the time to assure their manuscript submissions had such quality. I think this is a worthwhile manuscript to add to the literature.

REPLY: Thanks a lot

Reviewer 2 Report

Although this study is based upon a relatively small sample and there have been problems with drop-outs I still think it is a valuable contribution. The authors have used adequate questionnaires and statistics and the weaknesses have been described in a clear way so that the reader can build an estimation of the merit of the conclusions.

Author Response

Although this study is based upon a relatively small sample and there have been problems with drop-outs I still think it is a valuable contribution. The authors have used adequate questionnaires and statistics and the weaknesses have been described in a clear way so that the reader can build an estimation of the merit of the conclusions.

REPLY: We would like to express our appreciation for this reviewer’s comment.

Reviewer 3 Report

The main rationale for this research seems to be that the concept of marital stability hasn’t been considered yet among couples with acquired brain injury (ABI). However, the study’s aim seems to be investigating the impact of some clinical and demographic variables on marital stability, instead of the impact of marital relationship on quality of life of patients and their spouses. In the Introduction section, I miss a clear argumentation on the authors’ hypothesis, based on previous research, and on why this research is important.

Authors conclude to have demonstrated that some demographic and clinical features may have protective effect against marital instability in couples with ABI spouses. Lack of representativeness due to the small sample and the inclusion/exclusion criteria is acknowledged in the discussion section. However, I think that the most relevant limitation regarding representativeness (and then interpretation of data) consists in a possible selection bias. Indeed, patients who agreed participating the study (55% of the initial sample) might differ from those who refused or didn’t return the questionnaires (45%) for demographic characteristics or even for personal features that, as a latent variable, might impact both the desire to participate in the study and marital adjustment. Moreover, among those who agreed participating, 17 died before follow-up evaluation and 4 couples divorced. These aspects significantly limit the conclusions of the paper.

Introduction:

In the Introduction section (Line 41-44) they state: “The spouse often starts to neglect the couple relationship in favor of a care-relationship in which he/she takes care only of the daily life of the partner (washing him, dressing him, taking him to the bathroom, changing his diaper) and the management of his/her health status (physiotherapy, medication, medical examinations).” A reference should be provided in support of this statement.

In the Introduction section (Line 45-47) they state: “Moreover, the caregiver is also engaged in taking full charge of children, house, and work, whereas love comes to be replaced by other feelings mostly associated with the role of protection and care.” A reference should be provided in support of this statement.

In the Introduction section (Line 52-53) they state: “Couple stability is highly dependent on the stress level as perceived by the caregiver”. They need to provide a reference for this.

Material and Methods

In the Material and Methods section (Line 99): they need to change “:” with “)”.

In the Material and Methods section (Line 100-101) they state: “DAS is one of the most widely used tools in clinical and research to evaluate the functioning of the couple”. They need to provide a reference for this.

They should specify the DAS and FRI items’ characteristics (i.e. Likert scale and number of points).

Discussion

In Line 172-173 they state: “We have demonstrated that there are some demographic and clinical factors that may have a protective effect against marital instability”. I find it is a very confident statement, considering that the study cannot demonstrate a direct causal relationship between the variables.

In Line 176-177 they state: “we showed that couples enrolled in this study have found a new equilibrium to sustain their relationship”

In Line 181-182 they state: “The spouse of a patients who has experienced the threat of losing his partner generally understimates the negative aspects of his/her relationship”. They should provide a reference for this.

They should also provide adequate references for possible explanations provided in Lines 182-184 and line 184-186.

In Line 232-233 they state: “Religiosity can give hope, optimism, energy, security, and inner strength to solve problems”. They should provide a reference for this.

As regards the last part of the Discussion, where they list the limits of the research, they should indicate what I think is one of the most limitations of the study, that is a possible selection bias. Indeed, they only analyzed a little portion (self-selected) of the initial sample. It would be helpful to consider demographic information about couples that accepted to participate in the study but that were divorced at the follow-up would be interesting as well as about patients died before the follow-up evaluation.  

In the last part of Discussion, they should indicate future perspectives within the research line discussed in the paper.

Conclusions

I would suggest not to put a reference in the Conclusion section.

Author Response

The main rationale for this research seems to be that the concept of marital stability hasn’t been considered yet among couples with acquired brain injury (ABI). However, the study’s aim seems to be investigating the impact of some clinical and demographic variables on marital stability, instead of the impact of marital relationship on quality of life of patients and their spouses. In the Introduction section, I miss a clear argumentation on the authors’ hypothesis, based on previous research, and on why this research is important.

REPLY: Following the reviewer’s suggestion we modified the introduction in several parts to better present this topic.

Authors conclude to have demonstrated that some demographic and clinical features may have protective effect against marital instability in couples with ABI spouses. Lack of representativeness due to the small sample and the inclusion/exclusion criteria is acknowledged in the discussion section. However, I think that the most relevant limitation regarding representativeness (and then interpretation of data) consists in a possible selection bias. Indeed, patients who agreed participating the study (55% of the initial sample) might differ from those who refused or didn’t return the questionnaires (45%) for demographic characteristics or even for personal features that, as a latent variable, might impact both the desire to participate in the study and marital adjustment. Moreover, among those who agreed participating, 17 died before follow-up evaluation and 4 couples divorced. These aspects significantly limit the conclusions of the paper.

REPLY: We agree with this reviewer that the number of people who refused or didn’t return the questionnaires might affect the representativeness of our study, but we didn’t believe that this may prevent the possibility to conduct our study. However, we include a new limitation section discussing this reviewer’s suggestion. Please see line 286: “Third, a large number of patients fulfilling inclusion criteria refused to participate. Recruitment of substantially larger sample would mitigate concerns about the representativeness of our sample, although this kind of issue is usually when performing follow-up study with a large time window in this kind of neurological patients.”

Introduction:

In the Introduction section (Line 41-44) they state: “The spouse often starts to neglect the couple relationship in favor of a care-relationship in which he/she takes care only of the daily life of the partner (washing him, dressing him, taking him to the bathroom, changing his diaper) and the management of his/her health status (physiotherapy, medication, medical examinations).” A reference should be provided in support of this statement.

REPLY: Done. See Tooth et al. Caregiver burden, time spent caring and health status in the first 12 months following stroke. Brain Inj. 2005;19:963-974

In the Introduction section (Line 45-47) they state: “Moreover, the caregiver is also engaged in taking full charge of children, house, and work, whereas love comes to be replaced by other feelings mostly associated with the role of protection and care.” A reference should be provided in support of this statement.

REPLY: Done. See Marshall et al. Interventions to address burden among family caregivers of persons aging with TBI: A scoping review. Brain Inj. 2019;33(3):255-265. doi: 10.1080/02699052.2018.1553308

In the Introduction section (Line 52-53) they state: “Couple stability is highly dependent on the stress level as perceived by the caregiver”. They need to provide a reference for this.

REPLY: Done. See Kim D. Relationships between Caregiving Stress, Depression, and Self-Esteem in Family Caregivers of Adults with a Disability Occup Ther Int. 2017; 2017:1686143. doi: 10.1155/2017/1686143.

Material and Methods

In the Material and Methods section (Line 99): they need to change “:” with “)”.

REPLY: Done.

In the Material and Methods section (Line 100-101) they state: “DAS is one of the most widely used tools in clinical and research to evaluate the functioning of the couple”. They need to provide a reference for this.

REPLY: Done.

They should specify the DAS and FRI items’ characteristics (i.e. Likert scale and number of points).

 REPLY: Done.

Discussion

In Line 172-173 they state: “We have demonstrated that there are some demographic and clinical factors that may have a protective effect against marital instability”. I find it is a very confident statement, considering that the study cannot demonstrate a direct causal relationship between the variables.

 REPLY: Modified following reviewer’s suggestion.

In Line 176-177 they state: “we showed that couples enrolled in this study have found a new equilibrium to sustain their relationship”

 REPLY: This sentence has been modified.  

In Line 181-182 they state: “The spouse of a patients who has experienced the threat of losing his partner generally understimates the negative aspects of his/her relationship”. They should provide a reference for this.

 REPLY: Done. See Bonanno GA. Loss, trauma, and human resilience: have we underestimated the human capacity to thrive after extremely aversive events? Am Psychol. 2004;59(1):20–28.

They should also provide adequate references for possible explanations provided in Lines 182-184 and line 184-186.

 REPLY: Done. See Kitzmüller G, Asplund K, Häggström T. J Neurosci Nurs. 2012; 44(1):E1-13. doi: 10.1097/JNN.0b013e31823ae4a1

In Line 232-233 they state: “Religiosity can give hope, optimism, energy, security, and inner strength to solve problems”. They should provide a reference for this.

REPLY: Done. See Koenig HG. Religion, spirituality, and health: the research and clinical implications.  ISRN Psychiatry. 2012 Dec 16;2012:278730. doi: 10.5402/2012/278730.

As regards the last part of the Discussion, where they list the limits of the research, they should indicate what I think is one of the most limitations of the study, that is a possible selection bias. Indeed, they only analyzed a little portion (self-selected) of the initial sample. It would be helpful to consider demographic information about couples that accepted to participate in the study but that were divorced at the follow-up would be interesting as well as about patients died before the follow-up evaluation.  

REPLY: As said before we reformulated the limitation section following the reviewer’s suggestion.

In the last part of Discussion, they should indicate future perspectives within the research line discussed in the paper.

 REPLY: A new statement has been included following the reviewer’s suggestion

Conclusions

I would suggest not to put a reference in the Conclusion section.

 REPLY: Eliminated

Round 2

Reviewer 3 Report

I believe the manuscript has been significantly
improved and now warrants publication in Healthcare.